# Utilization of the Rat Tibial Nerve Transection Model to Evaluate Cellular and Molecular Mechanisms Underpinning Denervation-Mediated Muscle Injury

**DOI:** 10.3390/ijms25031847

**Published:** 2024-02-03

**Authors:** Christina Doherty, Monika Lodyga, Judy Correa, Caterina Di Ciano-Oliveira, Pamela J. Plant, James R. Bain, Jane Batt

**Affiliations:** 1Keenan Research Center for Biomedical Science, St. Michael’s Hospital, Unity Health Toronto, Toronto, ON M5B 1T8, Canada; christina.doherty@unityhealth.to (C.D.); monika.lodyga@unityhealth.to (M.L.); judy.correa@unityhealth.to (J.C.); caterina.diciano-oliveira@unityhealth.to (C.D.C.-O.); pamela.plant@unityhealth.to (P.J.P.); 2Division of Plastic Surgery, Faculty of Health Sciences, McMaster University, Hamilton, ON L8S 4L8, Canada; bainj@hhsc.ca; 3Department of Medicine, Temerty Faculty of Medicine, University of Toronto, Toronto, ON M5S 3H2, Canada

**Keywords:** gastrocnemius, fibro-adipogenic progenitors (FAPs), denervation, sensory protection, sural nerve, glial-cell-line-derived neurotrophic factor (GDNF), adipogenesis, primary cell culture, flow cytometry/FACS

## Abstract

Peripheral nerve injury denervates muscle, resulting in muscle paralysis and atrophy. This is reversible if timely muscle reinnervation occurs. With delayed reinnervation, the muscle’s reparative ability declines, and muscle-resident fibro-adipogenic progenitor cells (FAPs) proliferate and differentiate, inducing fibro-fatty muscle degradation and thereby physical disability. The mechanisms by which the peripheral nerve regulates FAPs expansion and differentiation are incompletely understood. Using the rat tibial neve transection model, we demonstrated an increased FAPs content and a changing FAPs phenotype, with an increased capacity for adipocyte and fibroblast differentiation, in gastrocnemius muscle post-denervation. The FAPs response was inhibited by immediate tibial nerve repair with muscle reinnervation via neuromuscular junctions (NMJs) and sensory organs (e.g., muscle spindles) or the sensory protection of muscle (where a pure sensory nerve is sutured to the distal tibial nerve stump) with reinnervation by muscle spindles alone. We found that both procedures reduced denervation-mediated increases in glial-cell-line-derived neurotrophic factor (GDNF) in muscle and that GDNF promoted FAPs adipogenic and fibrogenic differentiation in vitro. These results suggest that the peripheral nerve controls FAPs recruitment and differentiation via the modulation of muscle GDNF expression through NMJs and muscle spindles. GDNF can serve as a therapeutic target in the management of denervation-induced muscle injury.

## 1. Introduction

Traumatic peripheral nerve injury sustained in workplace, motor vehicle, and sporting accidents denervates muscle [1,2,3,4], resulting in immediate muscle paralysis and the rapid development of muscle atrophy. Muscle denervation injury is reversible, with near complete recovery of both its mass and function if timely reinnervation occurs, due to the muscle’s robust regenerative capacity [5]. If reinnervation is delayed, however, the muscle’s reparative ability is exhausted, and myofibers are replaced by fibrosis and fat [6,7,8,9,10]. Several neurotrophic factors, such as glial-cell-line-derived neurotrophic factor (GDNF), are produced by cells within the injured nerve (e.g., neurons and Schwann cells) and muscle [11,12], promoting nerve regeneration from the proximal injury site, but the rate is slow (at 1 to 3 mm/day [12,13,14,15]). Thus, following injury in the brachial plexus or upper arm in an adult, the regenerating nerve can take 2 years to reach the forearm and hand muscles, at which time, the atrophied muscles are replaced by fibrous connective tissue. Connective tissue is not contractile, destroys muscle architecture, and is non-receptive to reinnervation, thereby resulting in permanent physical disability and an impaired quality of life [4,5,6,7,8,9,10,11,12,13,14,15,16]. As these injuries predominantly impact young and working age populations, they also impart significant societal costs due to increased health resource utilization and lost productivity [1,17,18,19,20].

There are no therapies that can universally sustain long-term denervated muscle awaiting reinnervation. Surgical nerve repair, such as nerve transfer, which involves the transposition of a healthy motor nerve from the insertion site of a functionally less important muscle (sacrificing it) to the transected distal nerve stump of the denervated muscle, is standard of care [21]. Disability still results, but overall functionality is better maintained because the sacrificed muscle is not critical to prioritized movements. Sensory protection, where a pure sensory nerve is anastomosed to the muscle instead of a motor nerve, has also been trialed to sustain muscle awaiting reinnervation, and it has been found to partially mitigate muscle fibro-fatty degradation [22,23,24]. The absence of pharmacologic therapy to maintain denervated muscle awaiting reinnervation reflects the incomplete understanding of the cellular and molecular mechanisms underpinning the degradation, fibrosis, and fat infiltration of long-term denervated muscle.

Research over the past decade has established muscle-resident fibro-adipogenic progenitor cells (FAPs) as the cells that differentiate to adipocytes and fibroblasts to mediate the fibro-fatty replacement of traumatized muscle [25,26,27,28]. While the predominant research focus thus far has been to delineate FAPs intracellular signaling following muscle trauma [25,26,27,29,30,31,32], the upstream mediators and mechanisms controlled by muscle innervation that regulate FAPs recruitment, expansion, and differentiation remain incompletely understood. Peripheral nerves innervate muscle via both neuromuscular junctions (NMJs) [33], synaptic connections between motor neurons and muscle, where action potential and contraction are generated, and muscle sensory organs, such as spindles and Golgi tendons, responsible for sensing muscle stretch and regulating force production [23,34,35]. Conceivably, either upstream signals derived from disrupted NMJs and muscle spindles, or a lack of inhibitory signals released by intact NMJs and spindles, recruit and sustain FAPs in denervation injury.

In this study, we aimed to utilize the rat tibial nerve transection model to determine the mechanisms by which muscle innervation controls FAPs recruitment and differentiation. We demonstrate a changing FAPs phenotype in gastrocnemius muscle post-denervation, with an increasing propensity for adipocyte differentiation over time, and the mitigation of the FAPs response by immediate tibial nerve repair or the sensory protection of denervated muscle. We show that both procedures reduce denervation-mediated increases in the muscle expression of GDNF, and we identify GDNF as a peripheral nerve-regulated novel mediator of FAPs adipogenic and fibrogenic differentiation. GDNF may serve as a therapeutic target in the management of traumatic muscle denervation injury.

## 2. Results

### 2.1. Histologic Characterization of Gastrocnemius Muscle Subjected to Tibial Nerve Transection, Immediate Nerve Repair, and Sensory Protection

Progressive fibro-fatty infiltration and myofiber atrophy occurred in gastrocnemius muscle following tibial nerve transection (Figure 1A,B), in the absence of a significant inflammatory infiltrate (Figure 2). Immediate repair of the tibial nerve inhibited collagen and fat deposition, normalizing the muscle connective tissue content and rendering its morphology indistinguishable from that of healthy control muscle on histologic cross-sections. Sensory protection did not normalize the connective tissue content of denervated muscle. Progressive fibrosis and fat deposition occurred over time after tibial nerve transection in the sensory-protected muscle, but to a lesser extent than in denervated muscle, even at 12 weeks following nerve injury.

### 2.2. Adipogenic Differentiation of FAPs Isolated from Denervated Muscle

FAPs isolated from the 12-week denervated gastrocnemius muscle demonstrated increased transcript expression of markers of cellular proliferation and increased adipogenic differentiation compared to FAPs isolated from the control and 5-week denervated gastrocnemius muscle (Figure 3), supporting the premise that FAPs play a causative role in the irreversible deposition of connective tissue in long-term denervated muscle.

### 2.3. Sensory Protection and Immediate Tibial Nerve Repair Decrease FAPs Content of Denervated Gastrocnemius Muscle

Flow cytometric analysis was performed on rat gastrocnemius muscle 5-weeks after the denervation and repair interventions (sensory protection and immediate repair) to assess FAPs dynamics across the experimental groups (Figure 4). An increased number of FAPs was observed in the denervated gastrocnemius muscle relative to the gastrocnemius from the contralateral control limb. In contrast, the sensory-protected gastrocnemius, or muscle subjected to immediate tibial nerve repair, demonstrated a decreased FAPs content relative to the denervated muscle. Immunostaining for FAPs in the histologic cross-sections of gastrocnemius muscle at both 5 and 12 weeks post-tibial nerve transection also revealed an apparent decrease in the FAPs content in the muscle subjected to sensory protection compared to the denervated muscle (Figure 5). Immediate tibial nerve repair appeared to normalize the FAPs content of the gastrocnemius to that of the control muscle (Figure 5). Collectively, these data suggest that the mitigation and inhibition of muscle fibro-fatty infiltration observed with sensory protection and immediate tibial nerve repair, respectively, result, at least in part, from the negative regulation of FAPs expansion.

### 2.4. Denervation-Mediated Increase in Gastrocnemius Muscle GDNF Expression Is Inhibited by Immediate Nerve Repair and Sensory Protection

We sought to evaluate the gastrocnemius expression of GDNF, a neurotrophic factor whose dynamics have been previously reported to correlate with fibro-fatty infiltration in traumatically denervated muscle [11]. We reasoned that GDNF may play a causative role, as opposed to an associative role, in stimulating muscle fibro-fatty degradation after denervation and, thus, would be impacted by immediate tibial nerve repair and sensory protection. GDNF protein expression increased in the denervated gastrocnemius muscle (Figure 6). Sensory protection partially inhibited the increase in GDNF protein levels, and immediate tibial nerve repair normalized GDNF expression to baseline, comparable to the control gastrocnemius muscle. These data suggest that GDNF may play a mechanistic role in the mitigation and prevention of the fibro-fatty degradation of denervated muscle via sensory protection and immediate repair, respectively.

### 2.5. GDNF Is a Novel Positive Mediator of FAPs Adipogenic and Fibrogenic Differentiation

To investigate the impact of GDNF on FAPs adipogenic and fibrogenic differentiation, FAPs isolated from a naïve control rat were cultured with increasing concentrations of exogenous GDNF, and markers of proliferation and adipogenic and fibrogenic differentiation were assessed through RT-qPCR and immunostaining (Figure 7). GDNF induced FAPs adipogenic and fibrogenic differentiation in a dose-dependent manner, identifying GDNF as a novel positive mediator of FAPs adipogenesis and fibrogenesis.

## 3. Discussion

In this study, we aimed to utilize the rat tibial nerve transection model to determine the upstream mechanisms by which the peripheral nerve controls FAPs expansion and differentiation. We demonstrated the normalization and mitigation of denervation-induced FAPs increase in gastrocnemius muscle with immediate nerve repair and sensory protection, respectively, and we found that FAPs demonstrate progressive proliferative and adipogenic potential with increasing duration of denervation. Denervation induced a sustained upregulation of gastrocnemius GDNF expression, which was completely inhibited by tibial nerve repair with muscle reinnervation and was significantly decreased in the muscle provided (solely neurotrophic) support through sensory protection. We subsequently identified GDNF as a novel mediator of FAPs adipogenic and fibrogenic differentiation. Collectively, these data suggest that the peripheral nerve suppresses skeletal muscle GDNF expression through both the NMJ and the innervation of muscle sensory organs (e.g., spindles). This is lost with traumatic denervation, thereby promoting FAPs proliferation and differentiation, and ultimately resulting in irreversible muscle connective tissue infiltration and destruction.

FAPs’ cellular response differs substantially, dependent upon the type and chronicity of muscle injury. For example, in response to acute myotrauma, such as cardiotoxin injection in the mouse, FAPs are activated to massively expand in number by interleukin (IL)-4 secretion from a large influx of inflammatory cells, but by 10 days post-injury, FAPs return to basal levels, likely via apoptosis induced by macrophage Tumor Necrosis Factor (TNF)-α release [25,29,30]. In this scenario, FAPs exert a transient pro-regenerative role on muscle by stimulating satellite cell activation via secreted factors (e.g., IL-6, WNT family members, growth factors, and cytokines) and lay down a temporary extracellular matrix that supports regeneration [26,27,31,32]. In contrast, with traumatic denervation injury to muscle, there is a much slower trajectory of FAPs increase, and FAPs persist over weeks to months, coinciding with muscle fibro-fatty degradation, as we and others have previously shown [26,36] and as is observed in this study. Thus, the selection of the preclinical model is critical to understanding the biology of FAPs in the human disease or phenomenon of specific interest.

The tibial nerve transection model of the rat is an ideal model to study the mechanisms regulating traumatic denervation injury of muscle. Following nerve transection, the rat demonstrates normal activity and a minimally altered gait (only unable to spread its toes and fully dorsiflex the ankle), and the denervated gastrocnemius volume is adequate for multiple concomitant analyses from the single muscle of a single animal. This serves the ethical principles of minimizing animal morbidity and reducing the numbers used for experimentation. Rats are large enough to enable surgical nerve repair/grafting procedures, recapitulating those used in humans, and validated measures of rat toe spread and footprint length (SFI and SSI) serve as non-invasive indicators of gastrocnemius function in an alert and free-moving animal [37,38]. Most importantly, the tibial nerve transection model is an excellent mimic of human injury [5]. In contrast, genetic mouse models of muscle denervation may be confounded by concomitant muscle aging effects as the phenotype is manifested [39,40]. The small size of the mouse also limits the application of standard peripheral nerve surgical care, such as that used in humans, thus negatively impacting study translatability and suitability for determining the specific mechanisms of muscle traumatic denervation injury.

The stages of skeletal muscle sequelae following traumatic denervation and the capacity for muscle recovery with reinnervation are well defined [5]. Only the stage duration varies between species, with all stages being the longest in humans [5]. As stated previously, immediately after the denervation muscle is paralyzed, myofibers rapidly begin to atrophy, capillarity rapidly declines, satellite cell content transiently increases in a reparative effort, and there is a minimal inflammatory response. Denervation-induced muscle sequelae remain reversible at this early stage. In the rat, the muscle will fully regain mass and function if it undergoes good-quality reinnervation within 2 months of denervation [41,42,43]. In the absence of reinnervation, however, over the next 5 months in the rat, capillary content falls further, regeneration fails as satellite cell numbers decline, myofiber atrophy continues, NMJs degrade, spindles atrophy, and fibrosis and fat deposition occur. Reinnervation at this stage results in only partial restoration of the muscle mass and strength. Beyond 7 months in the rat, the denervated muscle culminates in a state where functioning muscle is irreversibly replaced by non-contractile connective tissue and is unable to receive reinnervation or respond to electrical stimulation [5,14,41]. In this study, the 5- and 12-week post-denervation time points were selected to represent both a fully reversible “short-term” time frame of muscle injury and a “long-term” time frame where irreversible fibro-fatty muscle destruction begins to occur, respectively.

This study also took advantage of the capacity of the model to enable microsurgical nerve repair techniques used in humans, to delineate upstream factors controlled by peripheral nerve innervation through the NMJ vs. the muscle sensory organs, which might modulate FAPs population expansion and differentiation. In the tibial nerve transection model, immediate microsurgical repair of the tibial nerve is known to result in muscle re-innervation via both the NMJ and sensory organs [23,44]. The motor axons innervate the NMJ. The sensory afferent axons, which compose portions of all peripheral nerves, including those traditionally referred to as “pure motor” nerves, innervate the spindles and Golgi tendons [23]. In contrast, the sensory protection of muscle has been previously reported to reinnervate muscle only via the muscle spindle [23,44]. Sensory protection has been utilized clinically to successfully sustain long-term denervated muscle awaiting re-innervation [22], although the insensate area of the limb generated by the transposition of the sensory nerve to the denervated muscle has reduced enthusiasm for, and utilization of, this repair method. The current study demonstrates the novel finding, that sensory protection, which solely reinnervates muscle sensory organs, mitigates muscle fibrosis and fat deposition by decreasing FAPs expansion and persistence in muscle. These data demonstrate that the upstream regulation of FAPs by the peripheral nerve can occur in the relative absence of NMJ interaction, highlighting the importance of the muscle sensory end organ when addressing therapeutics in muscle functional recovery, following denervation.

To determine the mechanism by which the peripheral nerve controls FAPs proliferation and differentiation, we sought to identify muscle factors previously reported to be regulated by both denervation and surgical nerve repair techniques. In a mouse model of acute (15 day) sciatic nerve transection, FAPs, derived from the denervated muscle, exhibited persistent intracellular STAT3 activation and increased IL-6 secretion, which promoted muscle fibrosis and myofiber atrophy [26]. However, the impact of nerve repair on FAPs was not evaluated, and the upstream mediators of the FAPs response were not identified. GDNF, a key neurotrophic factor and member of the Transforming Growth Factor β (TGFβ) superfamily, has been demonstrated to be persistently increased after nerve trauma in both denervated muscle and the distal nerve stump [11,12,45]. In an evaluation of neurotrophic factors (GDNF, BDNF, NT3, and NGF), the sensory protection of muscle decreased, and immediate nerve repair normalized, the denervation-induced increase in gastrocnemius GDNF transcript expression, suggesting that GDNF might regulate FAPs proliferation and differentiation [11]. The current study confirmed the significant and progressive upregulation of GDNF protein expression in the denervated muscle, with immediate tibial nerve repair decreasing (normalizing) the levels of GDNF expression in the gastrocnemius comparable to those of the control muscle. GDNF protein levels were also found to be suppressed by sensory protection, compared to the denervated muscle, at 12 weeks. This latter observation stands in contrast to reports where the sensory-protection-induced suppression of denervated muscle GDNF transcript expression was shorter lived, with mRNA levels increasing comparably to those of the denervated muscle by 12 weeks post-denervation [11]. This apparent discrepancy likely derives from the fact that changes in transcript levels do not necessarily reflect changes in protein levels. Importantly, it is the protein that impacts cell behavior. The normalization of GDNF protein levels with immediate tibial nerve repair and the significant, substantial decrease in GDNF protein levels in sensory-protected muscle observed in this study, suggest that GDNF may play a regulatory role in the fibro-fatty degradation of long-term denervated muscle.

GDNF is a known potent positive regulator of motor neuron survival, inducing neurite outgrowth and NMJ synapse formation [12,15,46]. Exogenous GDNF has been shown to exert positive effects on nerve regeneration and improve functional recovery in the context of experimental peripheral nerve injury [15]. GDNF is expressed in non-neuronal tissue and signals downstream through various receptors, including RET, NCAM-1, and syndecan, to mediate a multitude of effects [47]. In culture, exogenous GDNF activates hepatic stellate cells (the liver-resident mesenchymal cells that differentiate to myofibroblasts) to produce an extracellular fibrous matrix that culminates in hepatic cirrhosis [48]. In pre-clinical models of hepatic fibrosis, GDNF has been found to have varied effects, inducing cirrhosis in CCL4 and bile duct ligation models [48] but protecting against hepatic fibrosis in a high-fat-diet-fed model [49]. The reason for these discrepant results remains unclear. In skeletal muscle, GDNF has been demonstrated to inhibit satellite cell activation and proliferation via RET signaling, with an absence of impact on committed myoblasts [50]. The impact of GDNF on muscle FAPs has not been previously evaluated. In the current study, exogenous GDNF was found to induce the adipogenic and fibrogenic differentiation of FAPs, isolated from healthy gastrocnemius muscle and cultured in vitro, in a dose-dependent manner. When considered collectively, the data presented here suggest that the peripheral nerve suppresses skeletal muscle GDNF expression through both the NMJ and the innervation of muscle sensory organs (e.g., spindles), which is lost with traumatic denervation, thereby promoting FAPs differentiation and resulting in muscle fibro-fatty degradation.

The demonstration that GDNF stimulates FAPs adipogenesis and fibrogenesis, when considered in combination with the reports that GDNF is a negative regulator of satellite cell activation and proliferation, suggests that GDNF inhibition could serve as a therapeutic option to concurrently promote muscle regeneration and inhibit its fibro-fatty degradation following denervation. This, however, stands in contrast to the positive roles of GDNF in enhancing the regeneration of the severed peripheral nerve and maintaining the integrity of the NMJ, which is essential in preserving muscle receptivity to reinnervation. Given that the post-denervation changes in muscle take time to move from a “reversible” pathology to an “irreversible” pathology, it may be possible to time GDNF inhibition for the second phase of denervation injury (8 weeks/2 months and longer in the rat), which would enable the early positive impact of GDNF in neural regeneration prior to inhibiting its negative effects on muscle recovery. Alternatively, GDNF is capable of engaging multiple downstream receptors [47]. Future work requires the delineation of GDNF-mediated downstream signaling in FAPs, which may provide the opportunity to selectively inhibit GDNF’s effects on FAPs vs. the peripheral nerve.

There are limitations to this work. FAPs were isolated from the muscle via flow cytometry (FC)/FACS, and microheterogeneity in the FAPs phenotype with respect to cell surface markers has been shown to correlate with altered function [51]. In the current study, rat FAPs were sorted by negative selection for endothelial cells, hematopoietic cells (CD31/CD45), and satellite cells (VCAM-1), with Sca-1 as the positive FAPs selector [36]. However, due to the limited availability of conjugated and unconjugated primary antibodies that are effective for flow cytometry and that can recognize rat epitopes, other schema for FAPs isolation are not currently technically possible, limiting the evaluation of other possible/probable FAPs subsets.

Technical limitations also impacted Oil red O staining, which was used to identify neutral triglycerides. The short formaldehyde fixation and alcohol submersion in the method can dissolve some of the fat [52], but these steps were required to keep sections and cells adherent for staining. While the extent of fat deposition may therefore have been underestimated, the application of the same method across all experimental groups (control, denervation, immediate repair, and sensory protection) should have kept the relevant differences between groups apparent.

In summary, by using the tibial nerve transection model of the rat, with its capacity to enable clinically relevant microsurgical nerve repair, we demonstrate that the peripheral nerve can control FAPs proliferation and fibro-adipogenic differentiation, both via NMJ innervation and solely via muscle sensory organ innervation, engaging GDNF to mediate its effects. We demonstrate the novel finding that GDNF promotes FAPs adipogenic and fibrogenic differentiation, thereby contributing to the irreversible fibrosis and fat infiltration complicating long-term muscle denervation injury. GDNF might serve as a therapeutic target in situations of sustained denervation. The rat tibial nerve transection model is ideal to continue future studies of the regulation of muscle denervation injury, and the development and trialing of potential novel therapeutics.

## 4. Materials and Methods

### 4.1. Animal Model

A study on rats was conducted in accordance with the guidelines of St. Michael’s Hospital, Unity Health Toronto Animal Care Committee (ACC protocol 220).

Twelve female animals weighing 225 g (Lewis rats, Charles River Laboratories, Wilmington, MA, USA, strain code 004) were randomly assigned to 1 of 3 experimental groups (denervation, sensory protection, or immediate repair, as described by Zhao et al. [11], N = 4 per group). Briefly, the rats were maintained at a surgical plane of anesthesia with 2.5% isoflurane inhalation. One hindlimb was shaved and disinfected with 70% alcohol, followed by proviodine. An incision of approximately 1 cm in length on the lateral thigh, from the sciatic notch to the knee, was made, and the biceps femoris muscle was bisected with blunt-end scissors, revealing the sciatic nerve and its terminal branches (tibial, peroneal, and sural). The tibial nerve was identified and transected with micro-dissecting scissors 10 mm from its gastrocnemius insertion site. In the animals allocated to the denervation cohort, the proximal end of the transected tibial nerve was sutured to the anterior surface of the biceps femoris muscle to prevent spontaneous gastrocnemius reinnervation, using a 8-0 nylon suture. In the animals allocated to the immediate repair cohort, the transected tibial nerve was immediately re-anastomosed with a 10-0 nylon suture under the operating microscope. In the animals receiving sensory protection, the sural nerve was dissected out and transected with micro-dissection scissors, and the proximal end was sutured to the distal tibial nerve stump, using a 10-0 nylon suture under the operating microscope. The transected proximal tibial nerve ending was managed as above to prevent spontaneous gastrocnemius motor reinnervation. In all animals, hemostasis was ensured prior to closing the skin defect with 4-0 vicryl. The contralateral hindlimb remained unoperated, providing a healthy gastrocnemius muscle as an internal control for each animal.

The rats remained free roaming, with standard bedding and chow post-operatively. At serial times points (5 and 12 weeks) after tibial nerve transection, immediate repair, or sensory protection, the rats were anesthetized and sacrificed by a T-61 intracardiac injection. The gastrocnemius muscle was atraumatically dissected from the experimental and contralateral control hindlimbs, and each muscle was weighed and divided for downstream assays. Approximately two-thirds of the muscle was harvested in ice-cold saline for immediate processing via flow cytometry/FACS. The remaining tissue (one-third) was equally divided and either flash frozen in liquid nitrogen for RNA and protein extraction or frozen in liquid nitrogen-cooled isopentane for histology and immunohistochemistry. The latter samples were stored at −80 °C until processed.

### 4.2. Flow Cytometry/FACS

FAPs were identified and isolated via flow cytometry/FAC sorting, as previously described [36]. Briefly, a single cell suspension was generated from the gastrocnemius muscle using a combination of mechanical dispersion (mincing) and enzymatic digestion. The isolated cells were immuno-labeled with fluorophore-conjugated primary antibodies CD31::FITC (Abcam ab33858, Waltham, MA, USA endothelial cell marker), CD45::FITC (Biolegend 202205, San Diego, CA, USA, hematopoietic cell marker), VCAM::PE (Biolegend 200403 San Diego, CA, USA, satellite cell marker), and Sca-1 (Millipore sigma ab4336, Oakville, ON, Canada FAP marker) either self-conjugated to APC using a conjugation kit (Biotium, Burlington, ON, Canada 92307) or detected with goat anti-rabbit Alexa Fluor 647 (Invitrogen, A21244, Burlington, ON, Canada). SYTOX blue staining (Invitrogen, S34857, Burlington, ON, Canada) indicated cell viability. A sample analysis via flow cytometry for the characterization and quantification of cell type was conducted using a Fortessa X-20 benchtop cytometer (BD Biosciences, Mississauga, ON, Canada), and FAPs were isolated from the samples for cell culture using Aria III (BD Biosciences, Mississauga, ON, Canada). Single-stained and fluorescent-minus-one controls were used for each experiment, as previously described [36]. Data were acquired with FACSDiva Software (Beckton Dickenson, version 8.0.2, Mississauga, ON, Canada) and analyzed using FlowJo v.10 (Becton Dickinson, Mississauga, ON, Canada).

### 4.3. Primary Cell Culture

Cell culture medium contents can be found in the Appendix A. FACS-isolated FAPs were plated at a density of 5000 cells/cm^2^ in either 12-well (for RNA extraction) plates (Thermo Fisher, 130185, Burlington, ON, Canada) or 24-well (for immunocytochemistry) plates (Corning, 353047, Oakville, ON, Canada), and they were maintained at 37 °C with 5% CO_2_. FAPs were grown in basal growth media for 7 days until confluent, and then they were transferred to adipogenic differentiation media for an additional 7 days to stimulate adipogenesis. FAPs isolated for GDNF experimentation were maintained in basal growth media alone or with 15 ng/mL or 100 ng/mL GDNF (Bio-Techne, 512-GF, Minneapolis, MN, USA) for 14 days. Media were changed every 3 days.

### 4.4. Oil Red O Staining

A 0.35% (*w*/*v*) Oil red O (ORO) stock solution in 100% isopropanol was prepared by stirring overnight, passed through a 0.2 µm filter, and stored at 4 °C. An ORO working solution (60% ORO stock solution in ddH_2_O stirred for 20 min, passed through 0.2 µm filter) was prepared immediately prior to staining.

Isopentane-frozen gastrocnemius muscle was cut on a cryostat to generate 7 µm thick cross-sections. The sections were fixed in 4% paraformaldehyde (PFA, Electron Microscopy Sciences, 15710, Hatfield, PA, USA) for 10 min at room temperature, followed immediately by submersion in 60% isopropyl alcohol for 1 min. Slides were stained in ORO working solution for 12 min and then incubated in 60% isopropyl alcohol for 1 min before the placement of coverslips with water-soluble mounting media (Sigma, 1.08562.0057, Oakville, ON, Canada) for imaging

Cultured FAPs were fixed in wells with 10% Neutral Buffered Formalin (NBF, Millipore Sigma HT501128, Oakville, ON, Canada) for 5 min at room temperature. NBF was removed, and fresh 10% NBF was added for 1 h at room temperature. The cells were washed with 60% isopropanol and allowed to air dry (approximately 2–3 min), followed by incubation with 200 µL of ORO working solution for 10 min at room temperature. ORO was aspirated, and the wells were washed 5 times with ddH_2_O before imaging.

### 4.5. Picrosirius Red Staining

Isopentane-frozen gastrocnemius muscle was cut on a cryostat to generate 7 µm thick cross-sections. Slides were warmed to room temperature for 5 min, incubated with 100% ethanol for 1 min followed by 95% ethanol for 1 min, and washed 3 × 30 s in ddH_2_O. The slides were fixed in 4% PFA for 10 min and then washed twice for 2 min in ddH_2_O. The slides were stained with picrosirius red (PSR, Millipore Sigma, 365548-5G, Oakville, ON, Canada) for 3 min, immediately washed in acidified H_2_O for 15 s, and incubated at 65 °C for 15 min prior to the application of coverslips.

### 4.6. Immunostaining

#### 4.6.1. Platelet-Derived Growth Factor Receptor Alpha (PDGFRα) and Sca-1 Immunohistochemistry

Isopentane-frozen 7 µm thick gastrocnemius cross-sections were hydrated for 5 min in 1× PBS (Gibco, 10010-023, Burlington, ON, Canada), fixed in 4% PFA for 10 min at room temperature, and incubated for 90 min in IF blocking buffer (2% Bovine Serum Albumin, 5% FBS, 5% Goat Serum, 0.2% Triton-X, 0.1% sodium azide in phosphate-buffered saline). Slides were sequentially stained for Sca-1, PDGFR*a*, and laminin. Sections were incubated first with rabbit anti-Sca-1 (1:500, Millipore sigma, ab4336, Oakville, ON, Canada) overnight at 4 °C, detected with goat anti-rabbit Alexa Fluor 555 (1:500, Invitrogen, A21429, Burlington, ON, Canada) × 1 h, washed 3 × 5 min with PBST wash buffer (PBS + 0.05% Tween), and blocked with IF blocking buffer × 90 min. Slides were then incubated with rabbit anti-PDGFRα (1:250, Abcam, ab203491, Waltham, MA, USA) overnight at 4 °C, detected with goat anti-rabbit Alexa Fluor 647 (1:500, Invitrogen, A21244, Burlington, ON, Canada) × 1 h, washed 3 × 5 min with PBST wash buffer, and blocked with IF blocking buffer × 90 min. Final tissue immunostaining was with rabbit anti-laminin (1:500, Millipore Sigma, L9393, Oakville, ON, Canada) for 1 h at room temperature, detected with goat anti-rabbit Alexa Fluor 488 (1:500, Invitrogen, A11008, Burlington, ON, Canada), and washed 3 × 5 min with PBST wash buffer. Nuclei were labeled with Hoechst (1:10,000, ThermoFisher, 62249, Burlington, ON, Canada) for 5 min. All antibodies were diluted in 1× PBS + 0.05% Tween. Coverslips were mounted with antifade fluorescent mounting medium (Dako, S302380-2, Santa Clara, CA, USA), and slides were left overnight in the dark to dry at room temperature and stored at 4 °C prior to imaging.

#### 4.6.2. CD68/CD163 Immunohistochemistry

Isopentane-frozen 7 µm thick gastrocnemius cross-sections were hydrated for 5 min in 1× PBS, fixed in acetone for 10 min at −20 °C, air-dried, washed twice in 1× PBS for 4 min, and incubated for 60 min in IF blocking buffer (2% Bovine Serum Albumin, 5% FBS, 5% Goat Serum, 0.2% Triton-X, 0.1% sodium azide in phosphate-buffered saline). Slides were sequentially stained for CD68 or CD163 in addition to laminin. Sections were incubated first with anti-CD68 (1:200, abcam, ab213363, Waltham, MA, USA, M1 macrophage marker) or anti-CD163 (1:50, SantaCruz Biotechnology, SC-58965, Dallas, TX, USA, M2 macrophage marker) overnight at 4 °C; detected with goat anti-rabbit Alexa Fluor 555 (1:500, Invitrogen, A21429, Burlington, ON, Canada) or goat anti-mouse Alexa Fluor 555 (1:500, Invitrogen, A-21422, Burlington, ON, Canada), respectively, for 1 h; washed 3 × 5 min with PBS; immunostained with rabbit anti-laminin (1:500, Millipore Sigma, L9393, Oakville, ON, Canada) for 1 h at room temperature; and detected with goat anti-rabbit AlexaFluor488 (1:500, Invitrogen, A11008, Burlington, ON, Canada). Sections were washed 3 × 5 min with PBS. Nuclei were labeled with Hoechst (1:10,000) for 5 min, before being cover-slipped, left overnight in the dark to dry at room temperature, and stored at 4 °C prior to imaging.

#### 4.6.3. Perilipin-1(PLIN-1) Immunocytochemistry

Cultured FAPs were fixed with 4% PFA for 15 min at room temperature, washed 3 times with PBS, and then incubated with 100 mM Glycine in PBS × 10 min at room temperature to inactivate residual PFA. Cells were washed × 3 with PBS, incubated with 0.1% Triton-X in PBS × 20 min to permeabilize cells, washed × 2 with PBS, and blocked with 3% Bovine Serum Albumin (BSA) in PBS for 1 h at room temperature. Cells were incubated with anti-PLIN-1 (1:400, Abcam, ab3526, Waltham, MA, USA) overnight at 4 °C and detected with goat anti-rabbit Alexa Fluor 488 (1:400) × 1 h at room temperature. Nuclei were labeled with Hoechst (1:10,000) for 4 min. PBS was added to wells, and plates were stored at 4 °C in the dark until imaging.

### 4.7. RNA Isolation and RT-qPCR

Cells: RNA was isolated from cultured FAPs using an RNeasy Kit (Qiagen, 74104, Germantown, MD, USA), and DNAse treated as per the manufacturer’s instructions (Thermo Fisher, EN0521, Burlington, ON, Canada).

Skeletal Muscle: Flash-frozen gastrocnemius muscle was placed in liquid nitrogen in a clean, RNAse-free mortar and pestle and crushed into a fine powder. The powder was transferred to a 5 mL round-bottom tube on ice, and 1 mL of Trizol (Thermo Fisher 15596026, Burlington, ON, Canada) per 100 mg of tissue was added. Samples were homogenized for 3 × 30 s (Kinematica AG, Lucern, Switzerland) and then processed using a commercially available RNA extraction kit (Invitrogen, 12193555, Burlington, ON, Canada), with on-column DNAse treatment performed according to the manufacturer’s specifications (Invitrogen, 12185010, Burlington, ON, Canada).

RT qPCR: RNA quantity and quality were assessed using an Agilent 2100 Bioanalyzer (Agilent Technologies, Mississauga, ON, Canada). Only RNA with a RIN greater than 7 was used for analyses. Reverse transcription was performed on 1 µg DNase-treated RNA with Superscript III First-Strand Synthesis SuperMix (Invitrogen, 11752250, Burlington, ON, Canada) to obtain cDNA. qPCR was performed for genes of interest, with sets of primers designed and validated for specificity and efficiency. Gene amplification was performed with Power SYBR green PCR Master Mix (ThermoFisher 4367659, Burlington, ON, Canada) on a QuantStudio7 Flex Real-Time PCR System (ThermoFisher Scientific, Burlington, ON, Canada) with cycling parameters of 95 °C for 10 min, followed by 40 cycles of 95 °C for 15 s and 60 °C for 1 min. Samples were run in triplicate, and a “no-template control” was run for each primer pair. Data were analyzed using the comparative (∆∆CT) method [53]. Hypoxanthine-guanine phosphoribosyltransferase (HPRT), hydroxymethylbilane synthase (HMBS), and glyceraldehyde 3-phosphate dehydrogenase (GAPDH) variably served as housekeeping genes. Primer sequences are reported in Appendix A.

### 4.8. SDS-PAGE and Western Blot

Gastrocnemius total protein was extracted by homogenizing (Polytron PT 1200E, Kinematica AG, Lucern, Switzerland) the muscle in muscle lysis buffer [5 mM Tris/HCl, pH 8.0, 1 mM EDTA, 1 mM EGTA, 1 mM 2-mercaptoethanol, 1% glycerol, PMSF (1 mM) and leupeptin and aprotinin (10 μg/mL each)] three times for 30 s each, and homogenates were centrifuged at 1600× *g* for 10 min at 4 °C. The supernatant was cleared by centrifuging further for 10 min at 4 °C at 10,000× *g*. Protein lysates were quantified using the Pierce 660 nm Protein Assay Reagent (Thermo Fisher, 22660, Burlington, ON, Canada). Equal amounts of cytosolic protein lysate (25 µg) were separated on a 10% polyacrylamide gel, transferred to a nitrocellulose membrane, and stained with Ponceau-S (0.1% ponceau, 5% glacial acetic acid) to evaluate protein loading. Blots were blocked in 3% skim milk, blotted with anti-GDNF (1:1000, Invitrogen, PA5-88606, Burlington, ON, Canada), and detected with HRP-linked goat anti-rabbit secondary (1:5000, Thermo Fisher, 65-6120, Burlington, ON, Canada) using Clarity Western ECL Substrate (Bio-Rad, 1705060S, Mississauga, ON, Canada). The chemiluminescent signal was acquired on a Gel Doc EZ Imager (Bio-Rad Laboratories, Mississauga, ON, Canada), and GDNF expression was quantified on Image Lab 6.0.1 (Bio-Rad, Mississauga, ON, Canada), with experimental values normalized to the contralateral control.

### 4.9. Microscopy and Image Acquisition

Tissue histologic light microscopy images were acquired on a Zeiss Axioscan.Z1 (Zeiss, North York, ON, Canada) with a Hitachi 3-chip color camera (HV-F202SCL) and 20×/0.8 Plan Apochromat objective. Immunofluorescent imaging of tissue was performed on a Zeiss Axio Observer (Zeiss, North York, ON, Canada) with a Hammamatsu ORCA-R2 CCD camera and a 10×/0.3 EC plan-Neofluar objective. Images of cultured cells were obtained using the BioTek Cytation 5 Cell Imaging Multimode Reader (Agilent, Mississauga, ON, Canada) with a SONY IMX264 CMOS camera and a 10×/0.3 Plan Fluorite objective. Acquisition settings were kept constant across experimental conditions for each stain.

### 4.10. Image Quantification

FAPs PLIN-1 Expression and ORO Staining Quantification: Nuclei counts were determined with Fiji ImageJ software Version 1.54F, and determination of the percentage of cells positive for PLIN-1 or ORO was manually performed by a trained, blinded reviewer. PLIN-1 or ORO percent positive cells were reported as the number of PLIN-1- or ORO-positive cells/total number of cells in the microscopic field. A minimum of 25,000 cells were evaluated per sample.

Tissue PSR and ORO Staining Quantification: Fibrosis and fat contents of muscle histologic sections were determined using Halo v2.3 software. Tissue was manually outlined, and regions with folds, debris, and tissue edges were excluded prior to quantification. ORO and PSR tissue contents were calculated using the validated HALO Area Quantification v1.0 algorithm, as described in [54,55].

### 4.11. Statistical Analysis

GraphPad Prism software (version 10) was used for statistical analysis. Continuous data are reported as a mean +/− S.D. and were compared using students *t*-test or ANOVA followed by Tukey’s post-test analysis to compare multiple means, as appropriate. Statistical significance was assumed if *p <* 0.05.

## Figures and Tables

**Figure 1 ijms-25-01847-f001:**
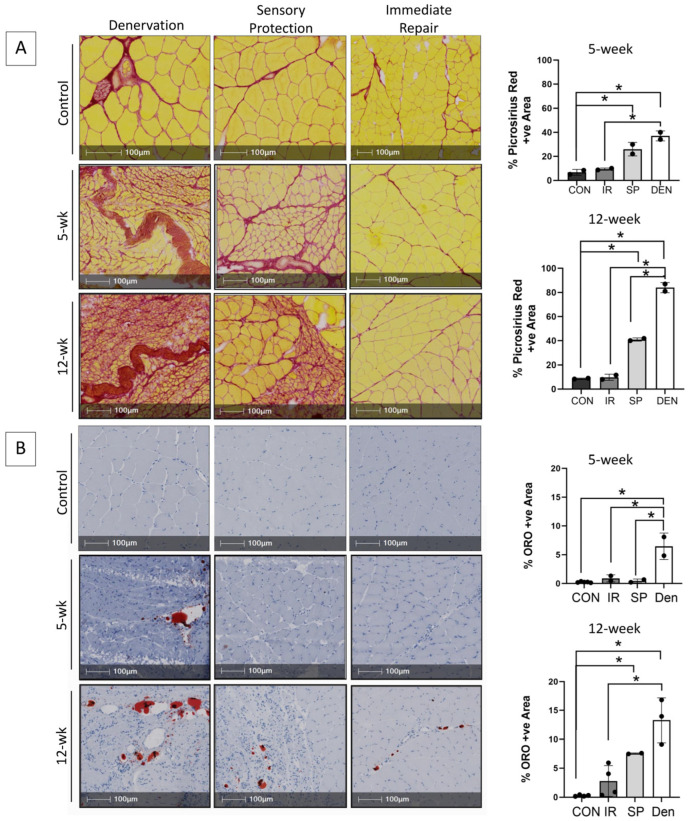
Immediate tibial nerve repair prevents and sensory protection decreases fibro-fatty degradation of gastrocnemius muscle in the rat tibial nerve transection model. Representative images of gastrocnemius histologic cross-sections stained with (**A**) picrosirius red delineating collagen (red) and myofibers (yellow) and (**B**) Oil red O (ORO) indicating neutral triglycerides (red); hematoxylin counterstains muscle purple. Morphometric analyses of histologic sections revealed progressive and significantly increased fibrosis and fat content in denervated (DEN) muscle over 12 weeks after tibial nerve transection compared to contralateral limb control (CON) muscle. Myofiber atrophy was also apparent. Immediate repair (IR) of the tibial nerve completely prevented gastrocnemius connective tissue deposition, with collagen and fat content being the same in control muscle at both the 5- and 12-week time points. Sensory protection (SP) significantly decreased gastrocnemius fibrosis and fat infiltration compared to denervated muscle, but it did not normalize connective tissue content to baseline comparative to healthy control and/or muscle subjected to immediate tibial nerve repair at the 12-week time point (* *p* < 0.05).

**Figure 2 ijms-25-01847-f002:**
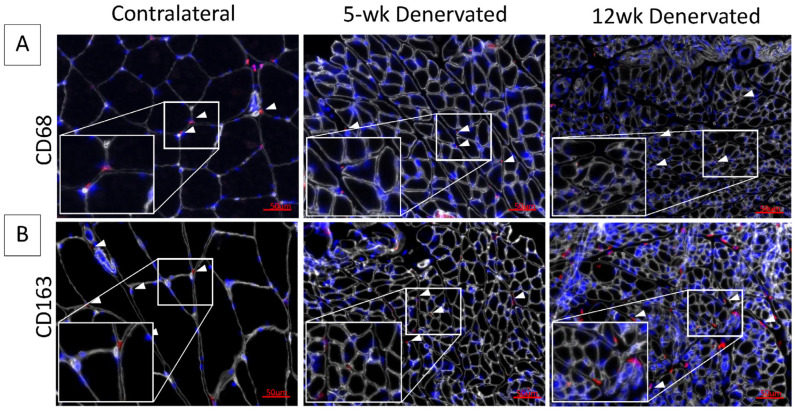
Immunostaining for inflammatory cell infiltrate in denervated gastrocnemius muscle. Representative images of contralateral control and denervated gastrocnemius histologic cross-sections immunostained for (**A**) CD68 (red) or (**B**) CD163 (red). Laminin staining (white) delineates myofibers. Nuclei are stained blue (Hoescht). White boxes show increased magnification of tissue sections. White arrows indicate hematopoietic cells. The inflammatory cell infiltrate appears minimal at 5 and 12 weeks (wk) post-denervation.

**Figure 3 ijms-25-01847-f003:**
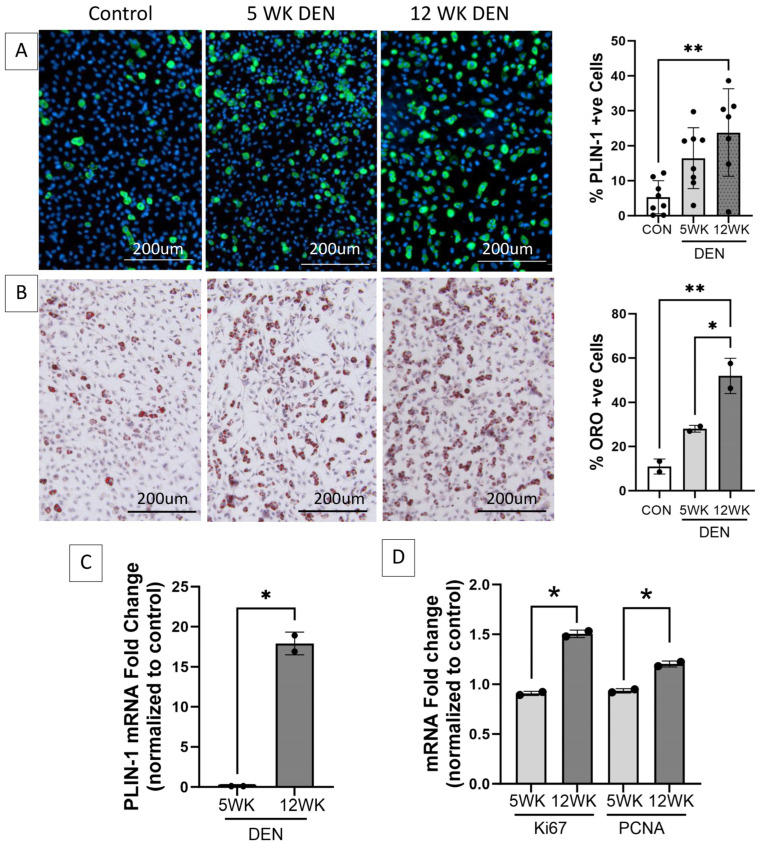
Increased adipogenic and proliferative capacity of fibro-adipogenic progenitors (FAPs) in denervated muscle. Representative images of FAPs isolated from control, 5-, and 12-week denervated gastrocnemius muscle and maintained in culture for 14 days (7 days growth media, 7 days adipogenic differentiation media) immunostained for (**A**) perilipin-1 (PLIN-1, green, a lipid droplet protein marker of mature adipocytes), where nuclei are blue (Hoechst), and (**B**) Oil red O (ORO, neutral triglycerides stain red), where nuclei are purple (hematoxylin). Morphometric analyses demonstrate significantly increased PLIN-1 expression and triglyceride production in FAPs isolated from 12-week denervated muscle, indicating increased adipocyte differentiation, compared to FAPs isolated from control and 5-week denervated gastrocnemius. (**C**) RT-qPCR for PLIN-1 and (**D**) RT-qPCR for proliferation markers Proliferating Cell Nuclear Antigen (PCNA) and proliferation antigen Ki67 demonstrate increased transcript expression in FAPs derived from 12-week, compared to 5-week, denervated gastrocnemius. HPRT and HMBS served as housekeeping genes (*, ** *p* < 0.05).

**Figure 4 ijms-25-01847-f004:**
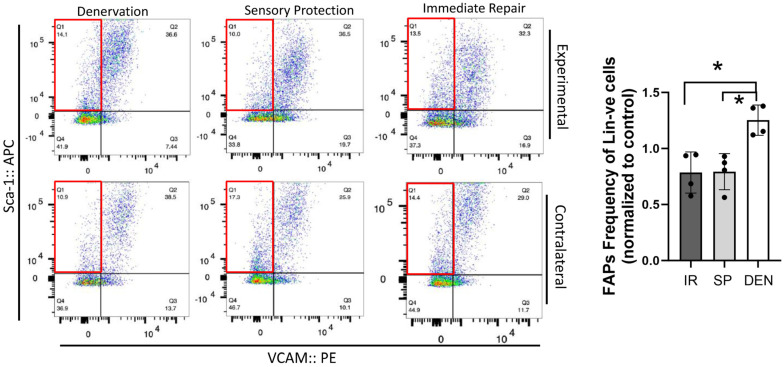
Flow cytometry reveals tibial nerve repair and sensory protection decrease FAPs content of denervated gastrocnemius muscle. Flow cytometric analysis of 5-week denervated, sensory-protected, and gastrocnemius muscle subjected to immediate tibial nerve repair, and contralateral control limb gastrocnemius. Single-cell suspensions were generated from muscle lysates. FAPs are identified as CD31-ve, CD45-ve, Sca-1+ve, VCAM-1-ve cells (red box—flow cytometry panels). FAPs content was significantly decreased in gastrocnemius subjected to immediate tibial nerve repair and sensory protection compared to denervated muscle (* *p* < 0.05, Lin-ve = CD31-ve/CD45-ve; DEN = denervation, SP = sensory protection, IR = immediate repair).

**Figure 5 ijms-25-01847-f005:**
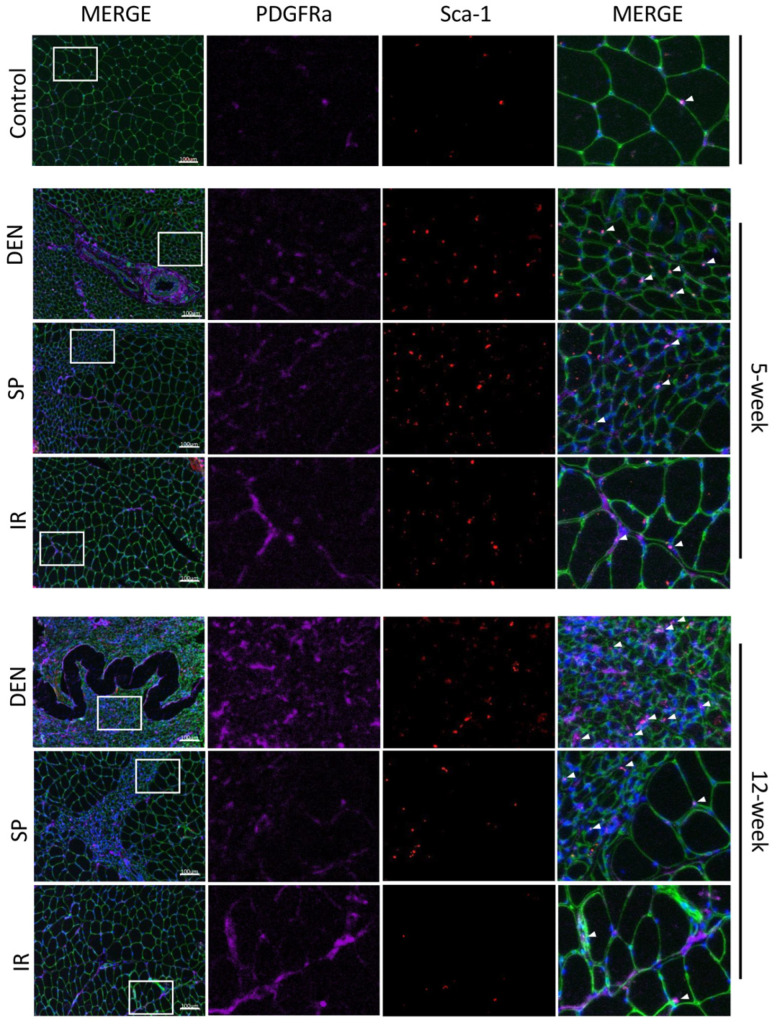
Immunostaining of histologic sections show tibial nerve repair and sensory protection decrease gastrocnemius FAPs content compared to denervated muscle. Representative images of gastrocnemius histologic cross-sections (scale bar = 100 µm) immunostained for Sca-1 (red) and platelet-derived growth factor receptor alpha (PDGFRα, purple). FAPs are identified by co-immunostaining of Sca-1 and PDGFRα (bright pink). Laminin immunostaining (green) outlines myofibers. Nuclei stain blue (Hoechst). White boxes in the left merged panels indicate tissue sections that are shown at higher magnification in the middle panels (Sca-1 immunostaining, PDGFRα immunostaining) and the right merged panels. FAPs are indicated with white arrows. Gastrocnemius FAPs content appears to increase to 12 weeks post-denervation (DEN) compared to both control (CON) muscle and muscle subjected to immediate nerve repair (IR). Sensory protection appears to partially mitigate FAPs expansion after denervation (CON = contralateral control; DEN = denervation, SP = sensory protection; IR = immediate repair).

**Figure 6 ijms-25-01847-f006:**
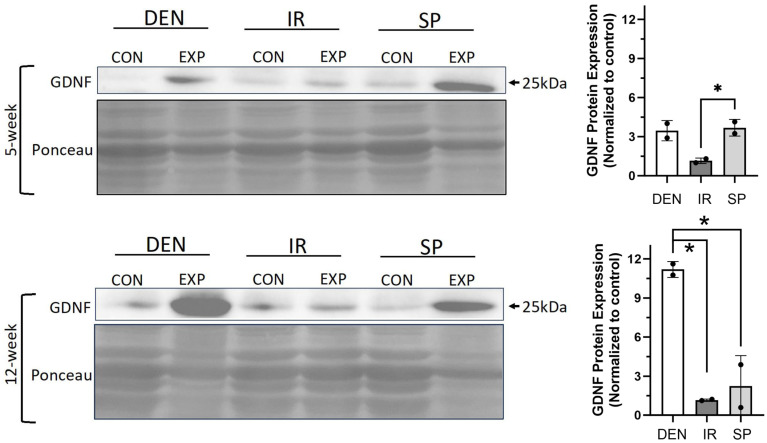
Sensory protection and immediate nerve repair decreased denervation-dependent increases in gastrocnemius GDNF expression. Representative Western blots of whole gastrocnemius protein lysates at 5 and 12 weeks after tibial nerve transection (DEN), tibial nerve repair (IR), sensory protection (SP) and contralateral control muscle. Denervation induced a sustained increase in GDNF protein expression, which was normalized to baseline in muscle subjected to immediate nerve repair. Sensory protection demonstrated decreased GDNF expression relative to denervated muscle at 12 weeks. Ponceau staining of Western blots served as loading control (* *p* < 0.05, EXP = experimental gastrocnemius, CON = contralateral control gastrocnemius).

**Figure 7 ijms-25-01847-f007:**
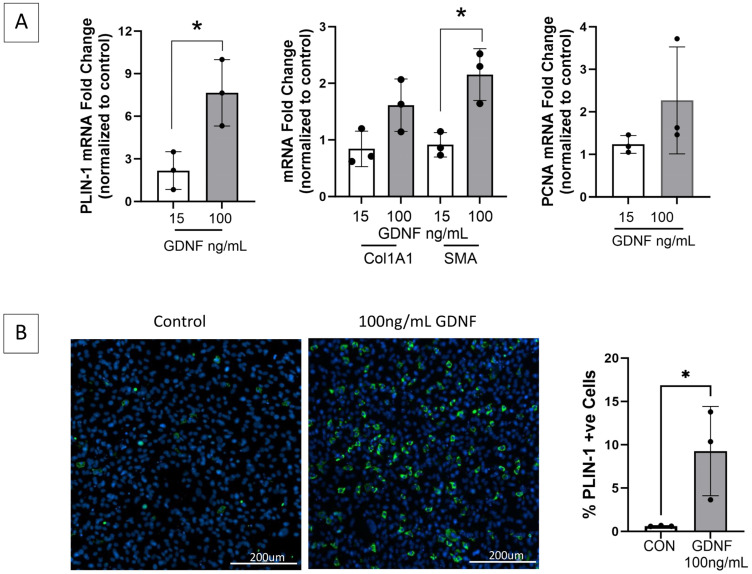
GDNF is a novel mediator of FAPs adipogenic and fibrogenic differentiation. (**A**) FAPs isolated from healthy gastrocnemius muscle and maintained in basal growth media alone or with GDNF 15 or 100 ng/mL for 14 days, demonstrate significantly increased transcript expression by RT-qPCR of perilipin-1 (PLIN-1, adipocyte marker) and fibrogenic differentiation marker smooth muscle actin (SMA), in a GDNF dose-dependent manner. Transcript levels of fibrogenic marker collagen 1A1 (Col1A1) and proliferation marker PCNA demonstrate increasing trends but are not significant. GAPDH and HPRT served as housekeeping genes. (**B**) Representative images of the GDNF treated- and control FAPs (growth media only) immunostained for PLIN-1 (green) demonstrate a GDNF dose-dependent increase in mature adipocytes (* *p* < 0.05; CON = control).

## Data Availability

Data are contained within the article.

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
