# Peer review of "Utilization of the Rat Tibial Nerve Transection Model to Evaluate Cellular and Molecular Mechanisms Underpinning Denervation-Mediated Muscle Injury"

_ijms, 2024, doi:10.3390/ijms25031847_

Round 1
Reviewer 1 Report
Comments and Suggestions for Authors
The experiment by Doherty, C. et al. about the evolution of denervation / reinnervation / neuroprotection in damaged nerves is exhaustive, appearing scientifically correct and well organized. I find interesting the nerve protection category in the experiment. As major comments I must say that some points the introduction seems reiterative and repeats some well-known facts. In addition the introduction, but particularly the abstract, should have their redaction revised to avoid reiteration and improve their appeal. For example, the abstract is no place to include the justification of your experiment.
There are also some minor comments:
- Abstract, lines 15-16. Mechanisms are understood.
- Abstract, line 20. FAPS of FAPs? Please, be consistent.
- Introduction. Line 33. Denervation occurs in traumatisms of peripheral nerves, or better said, when traumatisms affect peripheral nerves. I can´t reach the significance of the word “limb” in this sentence.
- Introduction, line 35. Include a comma after “reversible”. In line 37 also a comma before “however”. There are a lot of issues regarding the (lack of) commas.
- Materials, section 2.1, line 97. You should include here the number of rats. (XX female 225 Lewis rats…). Then, in discussion (line 440) you mention you want to minimize the number of experimental animals, but you don´t mention the number!
- Materials, section 2.1, lines 117-126. This paragraph has some confuse information. How much muscle you “harvested” (dissected seems a more appropiate word in anatomy or surgery)? Four thirds???
- Materials, section 2.2, line 128. Please, employ past tense to refer past events. FAPs were identified.
- Materials, section 2.4. I detect a methodological issue here. Did you know that formaldehyde fixation (that requires some kind of alcohol incubation) dissolves the fat? For this reason Oil Red O should be directly performed to cryostat sections if you want to see all the fat. In your case, you describe a very brief fixation and alcohol submersion (probably most fat was not dissolved), but, in any case, this is a limitation to demonstrate all the fat in the sample. I guess you have a methodological reason to perform such way, but you should briefly mention this limitation in discussion or, at least, in the proper materials & methods sections.
- Materials, section 2.7, line 234. Do you refer here to µg? By the way, in line 233 you want to say was or were used? (this issue is present in other sentences in the manuscript)
- Discussion, line 414. “Collectively, these data…”, or just ”Collectively, data…”
- Discussion, lines 419-422. You already mentioned this information (at least one time, may be more) in the introduction. The same for lines 470-475.
- Discussion, line 435. You can remove “with which”.
- Discussion, line 450. The stages of skeletal muscle sequelae following traumatic denervation are well defined is much clearer, as it identifies the subject (the stages) at the beginning…
- Discussion, lines 456-458. The same. Muscles devoid of innervation culminate….
- Discussion, line 459. In this case this is a complimentary sentence: with well defined time epochs.
- Discussion, line 494. Determine THE mechanism.
Comments on the Quality of English Language
The language is generally fine, but I can find several minor issues and strange expressions (some of them mentioned as minor issues). Authors are from Canada, there should be no issues at all. I must say that, at some points, sentence construction is confuse, not necessarily bad, but really difficult to read. I think a professional language review will decisively improve the text.
Author Response
We thank Reviewer 1 for their thorough and careful review of our manuscript. We have made several changes based on the suggestions, which we believe have substantially improved the manuscript. Responses to each comment are delineated below.
C1. The experiment by Doherty, C. et al. about the evolution of denervation / reinnervation / neuroprotection in damaged nerves is exhaustive, appearing scientifically correct and well organized. I find interesting the nerve protection category in the experiment. As major comments I must say that some points the introduction seems reiterative and repeats some well-known facts. In addition the introduction, but particularly the abstract, should have their redaction revised to avoid reiteration and improve their appeal. For example, the abstract is no place to include the justification of your experiment.
R1. We agree with the reviewer and have revised both the abstract and introduction substantially, to avoid reiteration as suggested. All changes are tracked in the uploaded, revised manuscript.
C2. There are also some minor comments:
- Abstract, lines 15-16. Mechanisms are understood.
- Abstract, line 20. FAPS of FAPs? Please, be consistent.
- Introduction. Line 33. Denervation occurs in traumatisms of peripheral nerves, or better said, when traumatisms affect peripheral nerves. I can´t reach the significance of the word “limb” in this sentence.
- Introduction, line 35. Include a comma after “reversible”. In line 37 also a comma before “however”. There are a lot of issues regarding the (lack of) commas.
R2. The manuscript has been reviewed by an editorial expert to address the problems with grammar and punctuation the reviewer highlights. All corrections have been made and are highlighted in the revised manuscript. In addition, FAPS has been changed to FAPs throughout, and line 33 has been revised to remove the reference to “limbs”.
C3. Materials, section 2.1, line 97. You should include here the number of rats. (XX female 225 Lewis rats…). Then, in discussion (line 440) you mention you want to minimize the number of experimental animals, but you don´t mention the number!
R3. We apologize for this oversight. Twelve rats were used and this number is now reported in the Methods section.
C4. Materials, section 2.1, lines 117-126. This paragraph has some confuse information. How much muscle you “harvested” (dissected seems a more appropriate word in anatomy or surgery)? Four thirds???
R4. We agree with the reviewer that the way the muscle dissection was described was confusing. The muscle was indeed divided into 3 pieces for different analyses. We have changed this to read, “Approximately two-thirds of the muscle was harvested in ice-cold saline for immediate processing via flow cytometry/FACS. The remaining tissue (one-third) was equally divided and either flash frozen in liquid nitrogen for RNA and protein extraction, or frozen in liquid nitrogen-cooled isopentane, for histology and immunohistochemistry.
C5. Materials, section 2.2, line 128. Please, employ past tense to refer past events. FAPs were identified.
R5. The tense has been changed to past tense.
C6. Materials, section 2.4. I detect a methodological issue here. Did you know that formaldehyde fixation (that requires some kind of alcohol incubation) dissolves the fat? For this reason Oil Red O should be directly performed to cryostat sections if you want to see all the fat. In your case, you describe a very brief fixation and alcohol submersion (probably most fat was not dissolved), but, in any case, this is a limitation to demonstrate all the fat in the sample. I guess you have a methodological reason to perform such way, but you should briefly mention this limitation in discussion or, at least, in the proper materials & methods sections.
R6. In the absence of a short period of fixation, we had difficulty with adherence of the tissue sections to the slides, even after drying. Fixation solved this problem. We have included a new paragraph in the section of the discussion dealing with this limitation. I provide here for your convenience.
“Technical limitations also impacted Oil red O staining, which was used to identify neutral triglycerides. The short formaldehyde fixation and alcohol submersion in the method can dissolve some of the fat [56], but was required to keep sections and cells adherent for staining. While the extent of fat deposition may, therefore, have been underestimated, application of the same method across all experimental groups (Control, Denervation, Immediate Repair, Sensory Protection), should have kept relevant differences between groups apparent”
C7. Materials, section 2.7, line 234. Do you refer here to µg? By the way, in line 233 you want to say was or were used? (this issue is present in other sentences in the manuscript).
R7. We have corrected “ug” , “um”, and “uL” to µg, µm, and µL throughout the manuscript
C8. - Discussion, line 414. “Collectively, these data…”, or just ”Collectively, data…”
- Discussion, lines 419-422. You already mentioned this information (at least one time, may be more) in the introduction. The same for lines 470-475.
- Discussion, line 435. You can remove “with which”.
- Discussion, line 450. The stages of skeletal muscle sequelae following traumatic denervation are well defined is much clearer, as it identifies the subject (the stages) at the beginning…
- Discussion, lines 456-458. The same. Muscles devoid of innervation culminate….
- Discussion, line 459. In this case this is a complimentary sentence: with well defined time epochs.
- Discussion, line 494. Determine THE mechanism.
R8. All of the comments above address incorrect grammar and redundancy in the manuscript. All of the grammatical errors are now corrected and substantial revisions have been made to address the redundancy. All of the changes are tracked, for the reviewer’s convenience.
C9. Comments on the Quality of English Language
The language is generally fine, but I can find several minor issues and strange expressions (some of them mentioned as minor issues). Authors are from Canada, there should be no issues at all. I must say that, at some points, sentence construction is confuse, not necessarily bad, but really difficult to read. I think a professional language review will decisively improve the text.
R8. As noted above, we had the manuscript reviewed by an individual with extensive editorial experience, and have corrected grammar and punctuation. We have also addressed any awkward language. We thank the reviewer for bringing this problem to our attention. We believe the manuscript is much improved.

Reviewer 2 Report
Comments and Suggestions for Authors
The Authors provided an interesting study on the explanation of the mechanisms by which the peripheral nerve regulates FAPs (Fibro-adipogenic Progenitor cells) expansion and differentiation. They developed the tibial nerve transection model in the rat and demonstrated a changing FAPs phenotype in gastrocnemius post-denervation with increasing adipocyte and fibroblast differentiation over time and mitigation of the FAPs response by tibial nerve repair. They have observed that both procedures reduced denervation-mediated increases in glial cell-derived neurotrophic factor (GDNF) in muscle and that GDNF promotes FAPs differentiation in vitro. This statement is characterized by a significant novelty. The methodology is adequate to resolve the problem as a hypothesis formulated precisely at the end of the Introduction section. The results are presented precisely and convincingly. The Discussion is interesting, and the study limitations are formulated honestly. The study fulfills the criteria of the basic study with the possible future utilization for clinical purposes.
The project undertaken by the Authors sounds. I have only some minor indications:
Line 95. Could you change “All animal work” to “The study on rats”, please?
Line 97 Could you change “Female 225g” to “Female animals weighing 225g, ”, please?
Line 99 Could you change “as described 11” to “as described by Zhao et al. [11]”, please?
Line 144 You may correct “are found” to “can be found”
Line 157 Please correct from “7um” to “7µm”, and please check for similar throughout the text (e.g. line 199 and the others)
If you use in the M&M section past tense, please keep it throughout as well in this section or subsections (e.g. line 128 …„FAPs are identified”…?).
Why not put (and make the short comment for) Figure S1 directly into the main text instead of in the supplemental material? Its content is convincing.
Are you sure that the style of citation in the text and the References list fulfills the MDPI criteria?
I believe, that after checking the paper's content editorially is it suitable for publishing in IJMS.
Comments on the Quality of English LanguageMinor English and editorial corrections are required.
Author Response
We thank Reviewer 2 for their critique of our manuscript and helpful suggestions. We have addressed their concerns in a point by point response below, and made the appropriate changes to the manuscript. We believe the manuscript is now much improved.
C1. The Authors provided an interesting study on the explanation of the mechanisms by which the peripheral nerve regulates FAPs (Fibro-adipogenic Progenitor cells) expansion and differentiation. They developed the tibial nerve transection model in the rat and demonstrated a changing FAPs phenotype in gastrocnemius post-denervation with increasing adipocyte and fibroblast differentiation over time and mitigation of the FAPs response by tibial nerve repair. They have observed that both procedures reduced denervation-mediated increases in glial cell-derived neurotrophic factor (GDNF) in muscle and that GDNF promotes FAPs differentiation in vitro. This statement is characterized by a significant novelty. The methodology is adequate to resolve the problem as a hypothesis formulated precisely at the end of the Introduction section. The results are presented precisely and convincingly. The Discussion is interesting, and the study limitations are formulated honestly. The study fulfills the criteria of the basic study with the possible future utilization for clinical purposes.
The project undertaken by the Authors sounds. I have only some minor indications
R1. We thank the reviewer for their positive assessment of our study and finding it worthy for publication in IJMS.
C2. Line 95. Could you change “All animal work” to “The study on rats”, please? Line 97 Could you change “Female 225g” to “Female animals weighing 225g, ”, please? Line 99 Could you change “as described 11” to “as described by Zhao et al. [11]”, please? Line 144 You may correct “are found” to “can be found”
R2. All these changes requested have been made. They are all tracked in the revised manuscript.
C3. Line 157 Please correct from “7um” to “7µm”, and please check for similar throughout the text (e.g. line 199 and the others)
R3. We have corrected “ug” , “um”, and “uL” to µg, µm, and µL throughout the manuscript.
C4. If you use in the M&M section past tense, please keep it throughout as well in this section or subsections (e.g. line 128 …„FAPs are identified”…?).
R4. The tense has been corrected to past tense to be consistent throughout the M&M section.
C5. Why not put (and make the short comment for) Figure S1 directly into the main text instead of in the supplemental material? Its content is convincing.
R5. Thank-you for this suggestion. We have now included this figure, as the new Figure 2, in the manuscript. The subsequent Figures have been appropriately re-numbered.
C6. Are you sure that the style of citation in the text and the References list fulfills the MDPI criteria?
R7. Thank-you for pointing out that the References were not cited as per MDPI criteria. This has now been corrected.
C8. Comments on the Quality of English Language. Minor English and editorial corrections are required.
R8. We had an individual with extensive editorial experience review the manuscript to correct grammatical and punctuation errors, and to address poorly formulated sentences. We feel that the manuscript’s “readability” is now vastly improved.